

# Brood size and sex ratio in response to host quality and wasp traits in the gregarious parasitoid *Oomyzus sokolowskii* (Hymenoptera: Eulophidae)

Xianwei Li, Liangting Zhu, Ling Meng and Baoping Li

School of Plant Protection, Nanjing Agricultural University, Nanjing, Jiangsu, China

## ABSTRACT

This laboratory study investigated whether the larval-pupal parasitoid *Oomyzus sokolowskii* females adjust their brood size and sex ratio in response to body size and stage of *Plutella xylostella* larval hosts, as well as to their own body size and the order of oviposition. These factors were analyzed using multiple regression with simultaneous entry of them and their two-way interactions. Parasitoids brood size tended to increase with host body size at parasitism when the 4th instar larval host was attacked, but did not change when the 2nd and 3rd instar larvae were attacked. Parasitoids did not vary in brood size according to their body size, but decreased with their bouts of oviposition on a linear trend from 10 offspring adults emerged per host in the first bout of oviposition down to eight in the third. Parasitoid offspring sex ratio did not change with host instar, host body weight, wasp body size, and oviposition bout. Proportions of male offspring per brood were from 11% to 13% from attacking the 2nd to 4th instar larvae and from 13% to 16% across three successive bouts of oviposition, with a large variation for smaller host larvae and wasps. When fewer than 12 offspring were emerged from a host, one male was most frequently produced; when more than 12 offspring were emerged, two or more males were produced. Our study suggests that *O. sokolowskii* females may optimize their clutch size in response to body size of mature *P. xylostella* larvae, and their sex allocation in response to clutch size.

## INTRODUCTION

Parasitoids make oviposition decisions depending on a variety of factors, including the quality of hosts and characteristics of ovipositing wasps (*Charnov et al., 1981*; *Van Alphen & Visser, 1990*; *Godfray, 1994*; *Ellers & Jervis, 2003*). Host quality for parasitoid development is often assumed to scale with host size and this pattern is true for idiobiont parasitoids, which arrest host growth at the time of oviposition (*Askew & Shaw, 1986*; *King, 1990*; *Otto & Mackauer, 1998*). This pattern, however, may not be universally applicable to koinobiont parasitoids, which allow the host to continue to develop after it has been parasitized (*Werren, 1984*; *Waage, 1986*; *Mackauer, Sequeira & Otto, 1997*; *Harvey, Sano & Tanaka, 2010*), and thus the future resources for developing koinobiont larvae can vary depending on the host

Corresponding author
Baoping Li, lbp@njau.edu.cn

age or development stage at the time of oviposition (*Harvey & Strand, 2002*; *Harvey, 2005*), though there are some koinobiont parasitoids still preferring larger body hosts (*King, 1989*; *Elzinga, Harvey & Biere, 2003*; *Kant, Minor & Trewick, 2012*). In addition to host quality, parasitoid oviposition decisions may also depend on characteristics of ovipositing wasps, including their body size and states, such as egg load, age, and nutrition states. The wasp size is frequently considered to be the main target for selection (*Godfray, 1994*; *Ode & Strand, 1995*; *Bezemer, Harvey & Mills, 2005*). It can affect fitness by influencing searching efficiency, longevity, or egg supplies (*Mayhew & Heitmans, 2000*; *Milonas, 2005*; *Jervis, Ellers & Harvey, 2008*; *Kasamatsu & Abe, 2015*). In addition, it may also influence the quality of host attacked, in that a small female wasp may not be able to use a large, good quality host that is more effective in defense against attacks (*Godfray, 1994*; *Henry, Ma & Roitberg, 2009*). Parasitoid state can affect many parasitoid behaviors (*Van Alphen & Visser, 1990*; *Roitberg & Bernhard, 2008*; *Harvey, Poelman & Tanaka, 2013*). For example, oviposition experience and physiological constraints, such as egg load, can influence parasitoid oviposition decisions (*Cloutier et al., 1991*; *Henneman et al., 1995*; *Sirot, Ploye & Bernstein, 1997*; *Ueno, 1999*; *Rosenheim et al., 2008*; *Dieckhoff et al., 2014*). Therefore, parasitoids may integrate the information provided by a variety of cues about hosts and themselves to make oviposition decisions.

*Oomyzus sokolowskii* (Kurdjumov) is a gregarious larval-pupal koinobiont endoparasitoid of the diamondback moth, *Plutella xylostella* (L.) (*Talekar & Shelton, 1993*; *Talekar & Hu, 1996*; *Silva-Torres et al., 2009*), which is one of major herbivorous pests on cruciferous crops (*Furlong & Zalucki, 2007*; *Zalucki et al., 2012*; *Li et al., 2016*). Because *P. xylostella* populations have developed resistance to numerous pesticides (*Liu, Tzeng & Sun, 1981*; *Ferré et al., 1991*; *Li et al., 2006*), biological control with natural enemies offers greater potential (*Talekar & Shelton, 1993*; *Delvare, 2004*; *Sarfraz, Keddie & Dosdall, 2005*). The parasitoid *O. sokolowskii* has been found parasitizing *P. xylostella* worldwide (*Wang et al., 1999*; *Nakamura & Noda, 2001*; *Silva-Torres et al., 2009*; *Li, Niu & Liu, 2014*), showing the potential for biological control of *P. xylostella* (*Talekar & Hu, 1996*; *Furlong, Wright & Dosdall, 2013*). For *O. sokolowskii* developmental performances in attacking *P. xylostella* larvae of different instars, two non-choice studies are not consistent in their conclusions, though the female wasp prefers mature larvae under choice circumstances (*Talekar & Hu, 1996*). While *Wang et al. (1999)* showed similar performances in the development time, number, and sex ratio of offspring in attacking host larvae of different instars, *Nakamura & Noda (2002)* showed that *O. sokolowskii* brood size increased with host body size when hosts for parasitism were confined to the prepupal stage. It would be interesting to examine whether or not parasitoid brood size varies as a function of host body size depending on host larval stage at parasitism. In addition, states of ovipositing wasps may influence oviposition decisions. It has been shown that older *O. sokolowskii* females declined in the progeny produced per female and the number of parasitoids emerged per host (*Silva-Torres, Barros & Torres, 2009*; *Sow et al., 2013*). However, scant attention has been paid to the effect of *O. sokolowskii* female body size on its oviposition behavior. Therefore, to fully understand *O. sokolowskii* oviposition strategy, we need studies involving both host quality and wasp traits.

This laboratory study aimed to answer the question: how do *O. sokolowskii* females make oviposition decisions on brood size and sex ratio in response to both host quality and their own traits? We examined combined effects of body size and age of *P. xylostella* larvae as well as body size and ovipositing order of the female wasp, by using mutiple regression models with simultaneous entry of these factors and their two-way interactions. Our study provides new data to the understanding of oviposition behavior in gregarious parasitoids.

## MATERIALS & METHODS

### Insects

*Plutella xylostella* was collected in the early spring 2014 from a pakchoi (*Brassica chinensis* L.) field on eastern suburb of Nanjing, Jiangsu, and thereafter maintained on potted pakchoi seedlings in insectary at 25 ± 1 °C, 60 ± 5% RH, and a photoperiod of 16:8 h (L:D).

*Oomyzus sokolowskii* was obtained from the Key Laboratory of Northwest Loess Plateau Crop Pest Management of Ministry of Agriculture of China, Northwest A&F University, Yangling, Shaanxi. It was maintained with *P. xylostella* 4th instar larvae as hosts in the insectary. A group of four host larvae was introduced into a 4 ml centrifuge tube, which was covered by 100 mesh plastic screen for ventilation and supplied with a piece of fresh pakchoi leaf as food. Then a female *O. sokolowskii* was released into the tube. After 24 h the larvae were removed from the tube and transferred individually to 10-cm-diameter Petri dishes containing excised fresh pakchoi leaves, which were replaced daily. The host pupae parasitized (different from normal ones by dark body color) were collected in groups in vials, where emerged parasitoid adults were maintained for one day to ensure fully mating. Naive females emerged within one day were used in the experiments.

### Oviposition behavior

*Plutella xylostella* 2nd, 3rd and 4th instar larvae were used as hosts. Each larva was individually exposed to parasitism after weighed (AL204-IC, with the measurement accuracy of 0.01 mg; Mettler Toledo, Columbus, OH, USA). After introduced into a 4 ml centrifuge tube containing a host larva, a female wasp was continuously observed until a bout of oviposition was completed. The wasp then was removed and immediately introduced into another tube containing a same-aged host larva for the second bout of oviposition, and then for the third. Three successive bouts of oviposition took ca. 5–6 h to be accomplished by a wasp, which was considered as a replicate. The number of replicates for the 2nd, 3rd, and 4th instar host larvae was 32, 43, and 34, respectively. After the three bouts of oviposition, the wasp was collected and measured in the right hind tibia length (HTL) under the microscope as a measure of body size. The parasitized host larvae were individually reared on excised pakchoi leaves in tubes until parasitoid offspring adults emerged. Emerged wasps from each host were counted as a measure of brood size, and their sexes determined by their antennae with long hairs on each segments in males and without the hairs in females. When the parasitized hosts were dead before wasps emerged, the hosts were dissected to count immature parasitoids as a measure of brood size, but their sexes were not identifiable.

## Data analyses

General linear mixed-effects models (LMMs) were used to analyze brood size (number of wasps emerged), which was transformed by logarithm before the analysis to satisfy the underlying assumptions. Generalized linear mixed-effects models (GLMMs) were used to analyze sex ratio (binary variable) with a binomial error. The subject wasp was included in the model as a random effects variable to control for the correlation between three oviposition bouts of each wasp. Host body weight, host instar (nominal), wasp size, the temporal order of oviposition (ordinal), and their second-way interactions were firstly included as fixed effects variables in the full model, then excluded from it when they did not have significant effects, using Type III Wald Chi-Square test to evaluate the significance of the terms by comparing the fit of full model to the fit of the reduced model without each of them. Orthogonal polynomial contrasts were used to detect trends (either linear or quadratic) in parasitoid performance with increasing bouts of oviposition. When there was a significant interaction between host instar and body weight, general linear models were separately used to detect the relationship between brood size and host body weight for each of host instars examined. To analyze the relationship between the number of males in a brood and brood size, the tree model was used to determine the threshold for splitting the brood size, of which the number of males in a brood changed as a step function. The mixed effects models were fitted by the laplace approximation using '**lme4**' version 1.1–10 (*Bates et al., 2015*). Data analyses were performed with R (*R Development Core Team, 2014*).

## RESULTS

Parasitoid brood size was affected by host body weight and instar interaction (Wald Chi-Square test, $\chi^2 = 6.30$, $P = 0.04$) and the order of oviposition ($\chi^2 = 18.01$, $P < 0.001$), but not by wasp HTL ($\chi^2 = 0.01$, $P = 0.94$). The regression analysis of brood size in relation to host body weight for different host instars showed no significant relations for the 2nd ($\chi^2 = 2.09$, $P = 0.14$) and 3rd ($\chi^2 = 0.90$, $P = 0.34$) instars. Brood size ranged from 4 to 21 wasps with an average of $9.0 \pm 0.3$ ($\pm$SE) in the 2nd instar hosts that varied in body weight from 0.01 to 0.03 mg, and 2 to 19 with an average of $9.1 \pm 0.2$ ($\pm$SE) in the 3rd instar hosts that varied in body weight from 0.03 to 0.14 mg (Figs. 1A and 1B). However, the analysis showed a marginally significant relation between brood size and body weight of the 4th instar hosts ($\chi^2 = 3.65$, $P = 0.048$), with a gradual increase by ca 20% in brood size with each increase of 0.1 mg in host body weight; brood size ranged from 4 to 14 wasps with an average of $9.3 \pm 0.3$ ($\pm$SE) across host body weight from 0.18 to 0.69 mg (Fig. 1C). Brood size did not change with ovipositing wasp HTL (Fig. 1D), but varied with the temporal sequence of oviposition, decreasing on a linear trend over increasing bouts of oviposition (slope $= -0.11$, SE $= 0.03$, $t = 4.24$) from 10 offspring adults emerged per host in the first oviposition bout down to 8 in the third (Fig. 1E).

Offspring sex ratio was not affected by host instar (Wald Chi-Square test, $\chi^2 = 0.89$, $P = 0.64$), host body weight ($\chi^2 = 0.16$, $P = 0.69$), wasp HTL ($\chi^2 = 0.01$, $P = 0.93$) and the temporal sequence of oviposition ($\chi^2 = 3.69$, $P = 0.16$). Average proportions of male offspring from attacking host larvae of different instars were 0.14 (SD $= 0.10$) for the

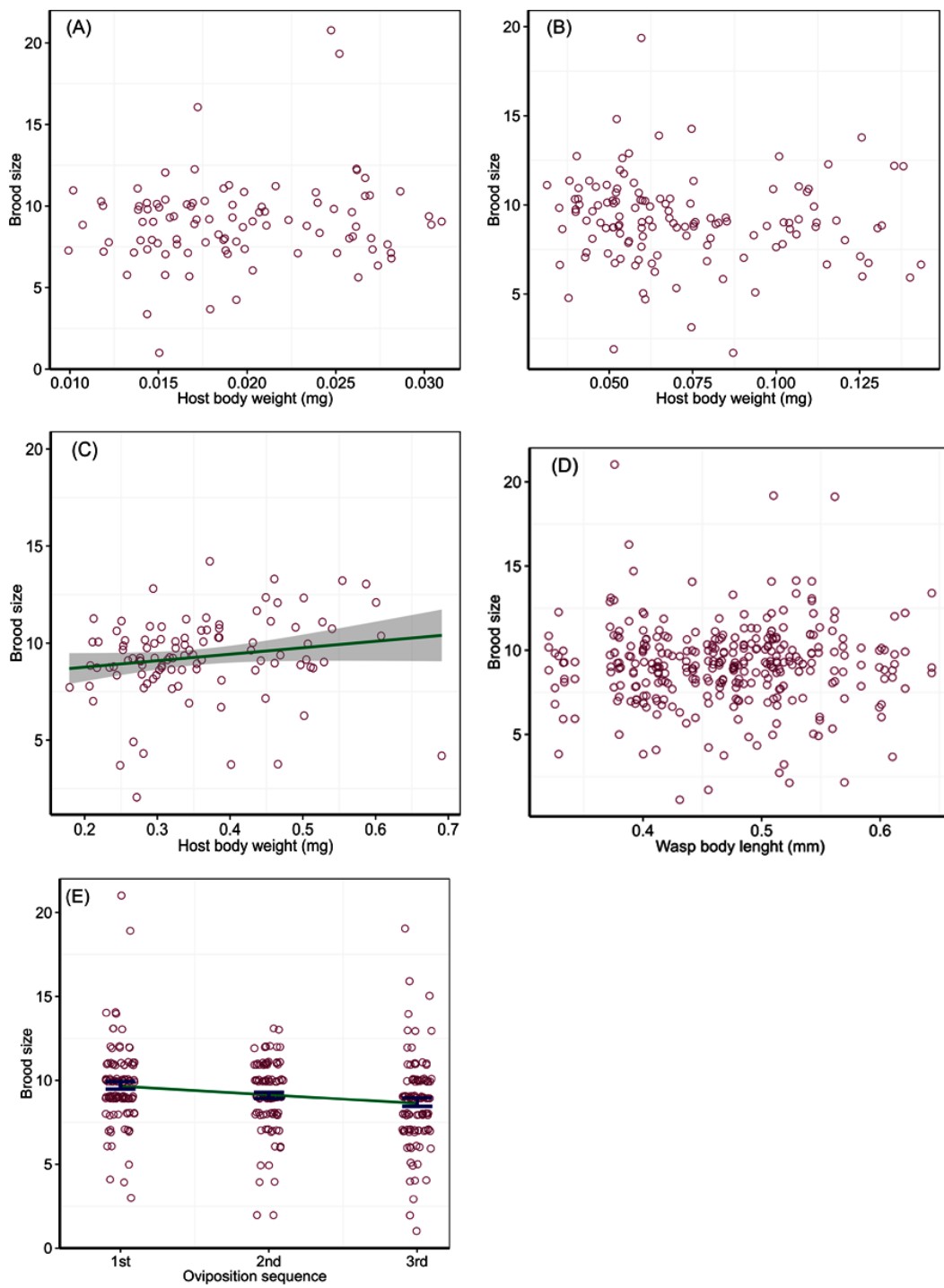

**Figure 1  Number of wasps emerged per host (brood size) in relation to host quality and wasp traits.**
Numbers of offspring parasitoids per host (brood size) in relation to host body weight of the 2nd (A), 3rd
(B) and 4th (C) instar larvae at oviposition, wasp body length (D), and the order of oviposition (E). The
solid line in (C) represents a linear relationship between brood size and host body weight with intercept
8.09 and slope 3.33, with a 95% confidence region (grey shade). The line in (E) represents a linear trend,
and the bars are mean and SE.
2nd and 0.14 (SD = 0.11) the 3rd instar larvae, with two extremely high sex ratios (male proportion > 0.5) for each of the two instars; the mean proportion of males was 0.13 (SD = 0.05) for the 4th instar hosts (Fig. 2A). Male proportion of offspring did not change with host body weight (Fig. 2B) and ovipositing wasp HTL (Fig. 2C), with a few extremely high sex ratios produced from very small hosts (body weight < 0.1 mg) or wasps (body length < 0.45 mm). Proportions of males for successive bouts of oviposition were 0.13 (SD = 0.09) for the 1st, 0.13 (SD = 0.06) the 2nd, and 0.16 (SD = 0.11) the 3rd, with two much higher sex ratios produced from the 1st and 3rd bouts of oviposition, respectively (Fig. 2D). The number of males per brood changed as a step function of brood size, which split at 12 with a significant difference between the two split groups ($F_{1,232} = 68.17$, $P < 0.001$). A brood fewer than 12 contained one male (mean = 1.2, SD = 0.49), while a brood of more than 12 had two males (mean = 1.68, SD = 2.24) with a few broods containing extremely numerous males (Fig. 2E).

## DISCUSSION

In this study, we showed that *O. sokolowskii* offspring brood size tended to increase with body size of *P. xylostella* 4th instar larvae, but did not change with that of the 2nd and 3rd instar larvae. The pattern exhibited in the 4th instar larval hosts is similar to that shown in prepupal hosts documented by *Nakamura & Noda (2002)*. We therefore assume that *O. sokolowskii* females may take the cues associated with body size of only older larval hosts when evaluating host quality to make the oviposition decision on clutch size. Host larvae close to pupation can provide more accurate information about host quality in terms of body size than younger ones do, because body size changes much less with growth in mature than younger larvae. Such evaluation of host quality can be advantageous for koinobiont parasitoids to attack older larval hosts, such as larval-pupal parasitoids. Female *O. sokolowskii* prefers to parasitize older than younger *P. xylostella* larvae (including prepupae) (*Talekar & Hu, 1996*). Though differential mortality rates during offspring development among different host instars may somehow potentially contribute to the pattern we observed in *O. sokolowskii*, such effect might be negligible because brood sizes were quite close to each other between host instars observed in our study, while the number of eggs (primary clutch size) are not different among these instars (*Nakamura & Noda, 2002*).

Body size is one of important attributes for female wasps because it is correlated with several components of the fitness (*King, 1987*; *Blackburn, 1991*; *King, 1993*; *Ellers & Jervis, 2003*; *Henry, Ma & Roitberg, 2009*). In the case of gregarious parasitoids, the fitness of a female parasitoid can be strongly correlated with not only the number of eggs laid per host but also the number of hosts attacked. Our study showed that *O. sokolowskii* brood size did not change with body size of ovipositing wasps. We therefore assume that larger *O. sokolowskii* females may realize their greater potential in reproduction by attacking more hosts instead of laying more eggs per host. Such strategy of clutch size allocation can be adaptive to exploiting clumped distribution of hosts. As the only host of *O. sokolowskii*, *P. xylostella* larvae tend to clump together in the early stage, disperse locally as they grow, and then aggregate again during pupation (*Chua & Lim, 1979*; *Furlong, Wright & Dosdall, 2013*).

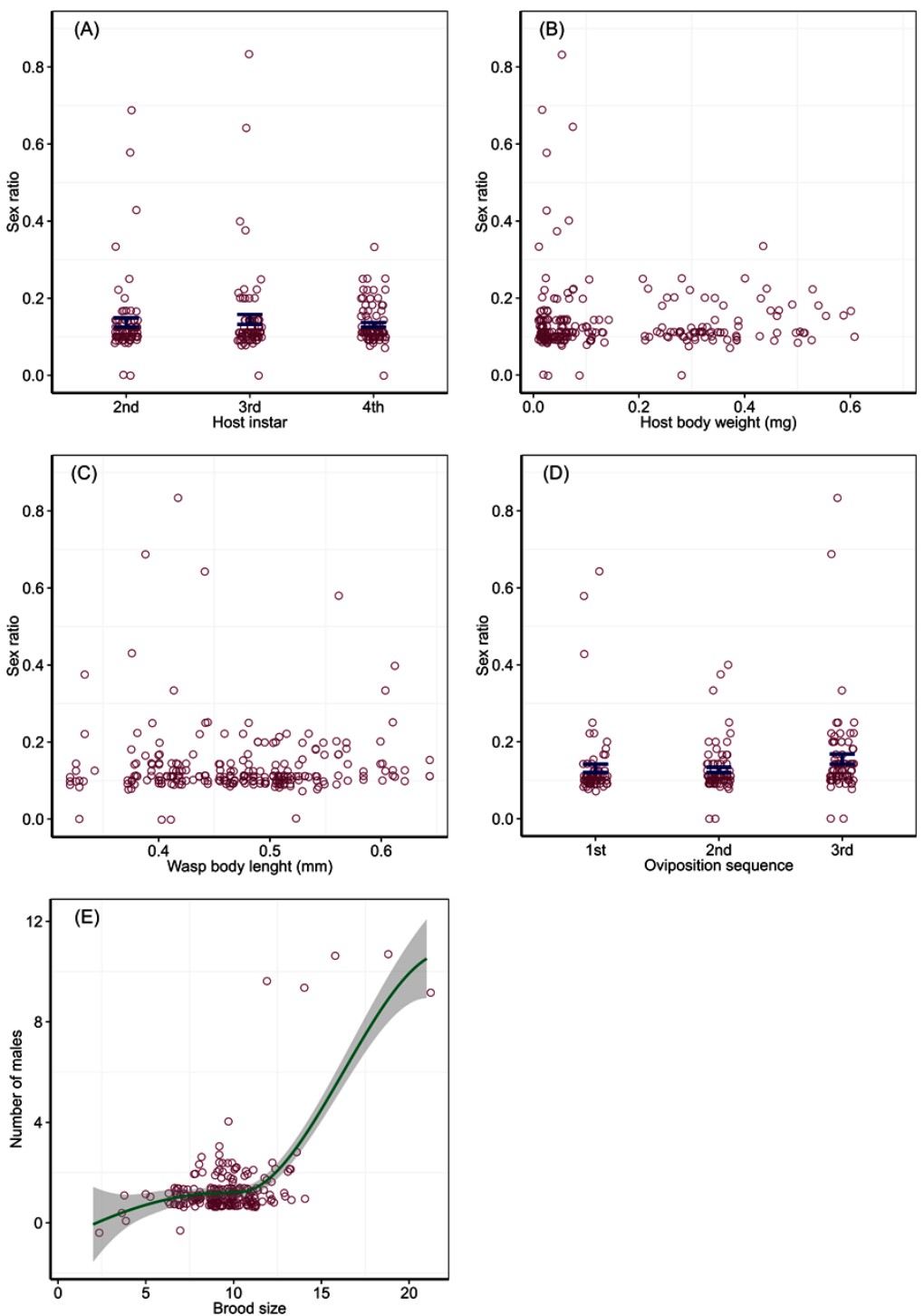

**Figure 2  Brood sex ratio in response to host quality and wasp traits.** Sex ratios in relation to host instar (A), host body weight (B), wasp body length (C) and bout of oviposition (D), and the relationship (E) between the number of males and offspring emerged (brood size) per host with a curve fitted by locally weighted polynomial method with a 95% confidence region (grey shade). Bars in (A) and (D) are mean ± SE.

For the performance of a slight, linear decrease in brood size across successive bouts of oviposition shown in our study, the most likely explanation is that egg expenditure on successive bouts of oviposition leads to a steady decline in number of eggs available to oviposit when a group of hosts are encountered in quick succession. Daily decrease in offspring production occurs in *O. sokolowskii* (*Wang et al., 1999*; *Li, Niu & Liu, 2014*), as well as in other gregarious parasitoids, such as in *Cotesia glomeratus* L. attacking the white butterfly *Pieris rapae* (L.) (*Ikawa & Suzuki, 1982*).

Our study did not show the influence of host size, host life stage, wasp body size and oviposition sequence on parasitoid offspring sex ratio. Invariable sex ratios in response to host age and size were previously observed in *O. sokolowskii* (*Wang et al., 1999*; *Nakamura & Noda, 2002*; *Silva-Torres et al., 2009*). The lack of sex allocation response to changes in host size in *O. sokolowskii* may exclude widely held Charnov's host quality hypothesis as an explanation. Charnov's hypothesis (*Charnov et al., 1981*) was developed specifically for solitary parasitoids, in which offspring body size is strongly correlated with host body size. Host size for many idiobiontic solitary parasitoids is thought to more strongly influence female than male fitness (*Werren, 1984*; *Charnov & Skinner, 1985*; *King, 1993*). In the case of gregarious parasitoids, however, the fitness of the adult wasp is determined primarily by the clutch size and possibly the sex composition (*Griffiths & Godfray, 1988*; *Hardy, Griffiths & Godfray, 1992*; *Godfray, 1994*; *Ode & Heinz, 2002*). Maintaining constant, female-biased sex ratios in response to variation in host quality and wasp traits in *O. sokolowskii* suggests that this parasitoid may be able to adjust offspring sex ratios. Female-biased sex ratios are most frequently exhibited in gregarious parasitoids, where there is a high probability the mating occurs between siblings and that brothers compete together for mates (*Godfray, 1994*; *Ode & Hunter, 2002*; *Ode & Hardy, 2008*). Such observations can be explained by *Hamilton*'s (*1967*) theory of local mate competition (LMC), which predicts that a single female should lay only enough males to fertilize all her daughters. In the case of *O. sokolowskii* where sib-mating is common among offspring from a host, foundresses are expected under LMC to adjust the number of males in response to brood size. Our study showed that *O. sokolowskii* produced one male in most of broods under 12 wasps, and two males in broods over 12, except a few largest broods with extremely numerous males from attacking very small host larvae. We suspect that these few broods were consequences of constraints under non-choice circumstances in the experiments. When these extreme data points are excluded, our study is the same as *Nakamura & Noda (2002)* in finding that only one male was produced when the brood contained fewer than nine offspring and 2–4 males when it contained more than nine offspring.

## ACKNOWLEDGEMENTS

We thank Yuanxing Sun for providing *O. sokoloskii* and Lintao Li for the help in the experiments, and the two reviewers for providing helpful comments on improving the manuscript.

### Funding

This research was supported by the Natural Science Foundation of China (NSFC-31570389). The funders had no role in study design, data collection and analysis, decision to publish, or preparation of the manuscript.

### Grant Disclosures

The following grant information was disclosed by the authors:
Natural Science Foundation of China: NSFC-31570389.

### Competing Interests

The authors declare there are no competing interests.

### Author Contributions

- Xianwei Li conceived and designed the experiments, performed the experiments, analyzed the data, contributed reagents/materials/analysis tools, wrote the paper, prepared figures and/or tables.
- Liangting Zhu performed the experiments, contributed reagents/materials/analysis tools.
- Ling Meng conceived and designed the experiments, analyzed the data, contributed reagents/materials/analysis tools, reviewed drafts of the paper.
- Baoping Li conceived and designed the experiments, analyzed the data, contributed reagents/materials/analysis tools, wrote the paper, reviewed drafts of the paper.

### Data Availability

The raw data has been supplied as Data S1.

### Supplemental Information

Supplemental information for this article can be found online at http://dx.doi.org/10.7717/peerj.2919#supplemental-information.

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
