# Peer review of "Brood size and sex ratio in response to host quality and wasp traits in the gregarious parasitoid Oomyzus sokolowskii (Hymenoptera: Eulophidae)"

_PeerJ, doi:10.7717/peerj.2919_

## Round 0.1 · original submission · Minor Revisions

I agree with the reviewers that your manuscript offers a valuable contribution and should be acceptable with revision. Please pay close attention to the comments of the reviewers in making your revision and in preparing your rebuttal.

·

Basic reporting

The manuscript presents information on the effect of host size on parasitoid brood size and sex ratio. The text is comprehensible but needs English revision. We also strongly suggest that the language be reviewed by a native English speaker. The appropriate literature is presented and the authors stated clearly the objectives.

Figures are clear but some embedded charts lack legend.

Experimental design

The experiment was carried out properly, but number of replications should be included. It is clear that the authors adopted a no-choice type of experiment. Therefore, this should be well explored in the Results and Discussion sessions (see below)

Validity of the findings

The authors mentioned in the discussion that Silva-Torres et al. (2009) found contrasting results. Silva-Torres et al. (2009) observed that 2nd- and early-4th-instar Plutella xylostella larvae produced parasitoids of similar sizes whereas in the submitted manuscript the authors found that larger number parasitoids were produced in larger hosts. We believe this issue should be better explored in the manuscript. Also, for the explanation on lines 147-156, the authors should also address that the experiment was carried with no choice and each wasp was given one larva to oviposit. By not having a choice, will ever smaller hosts (“poor hosts”) be chosen if both smaller and larger larvae were available? Although the authors did not test that, they should discuss this issue.
Taking a look at the raw data, one could observe that there was no emergence of insects from almost 30% of the broods. This was not addressed and may be an issue to be discussed.

Additional comments

The findings were very interesting but the authors should consider that the experiment used a no-choice approach. This approach may limit some conclusions and should be addressed. Also, explore more deeply the contrasting results such as those reported by Silva-Torres et al. (2009).

Reviewer 2 ·

Basic reporting

The authors the studied oviposition strategy (brood size and offspring sex ration) of an important larval parasitoid Oomyzus sokolowaskii of worldwide pest of crucifers Plutella xylostella. Paper conforms to the journal standard however it needs improvement on the clarity. Objectives of the research are clear but need some detail background information in the introduction. For example – Line # 39 “it may not be applied to koinobiont parasitoids …” but there are quite a few references available such as Fitness gain in koinobiont … (http://www.sciencedirect.com/science/article/pii/S1226861511001129).
Ln # 73 “However, scant attention …” quite a few reference such as Body size and fitness relation http://link.springer.com/article/10.1007/s10526-012-9452-4; http://www.ncbi.nlm.nih.gov/pubmed/19504128
I don’t want the authors to cite all these references but few examples of work in these areas. Author needs to provide good justification of this work, with proper references, in the introduction.
Another issue is use of “adjusting clutch size or brood size” ; this can’t be claimed from data presented here because the authors didn’t dissect the host to check the number of eggs laid by female wasps. It could be possible that not all the eggs laid by females hatched or larvae developed due to insufficient resources.
Methodology is inadequate to repeat the experiment if someone wants to. Few places texts are not clear and need good English language edits for clarification. Number of wasps used in this research, hosts used in the sequence such as hosts offered one by one or all offered at a time, what host instar offered during this or just random, this makes insufficient information to repeat the experiments. Also Ln # 85 – no need to use both common name and scientific names again in the M&M. Ln # 90-91 – what age or stage of P. xylostella larvae used here? It’s important to show mean and SE or SD in the results.
Ln # 153-156 – this is not always true there are wealth of references on aphids choose larger hosts than smaller. For example – one of the above suggested references.
Ln # 58-59 – “ The results …as suggested above we can’t conclude this without dissecting the hosts to confirm the number of eggs laid by the wasp..
Citation – authors used names of three authors rather et al., I am not sure if this is conforms to the journal format.
I would recommend discussion needs to be revised for clarity and minimise the speculations. Figures are adequate and presented well.

Raw data - no data for laraval instar

Experimental design

The study suits the scope of the journal and the objectives of the research are clear and relavant.
As mentioned earlier, methodology is inadequate if someone wants to repeat the experiments.

Validity of the findings

Data analysis sounds ok. Provide means and SD or SE in the results. some speculations need to be removed as rephrasing the texts in the discussion.

Additional comments

As above

---

## Round 0.2 · Minor Revisions

I apologize for this delay in getting back to you. We were waiting to hear from a second reviewer. However, I think the comments from the first reviewer will be sufficient. Would you please address the points from this review. I am hopeful that this will not take to much effort on your part and we can move forward with acceptance and publication

·

Basic reporting

The text was greatly improved and is much clear. I still have some minor suggestions as follow:
Lines 63-64 – References should be presented in chronological order.
Line 72 – Consider Nakamura & Noda (2002) instead of Nakamura and Noda (2002)
Lines 97-98 – “...females were individually provided with four 4th instar host larvae to parasitize for 24 h in a 4 ml centrifuge tube...” In fact the females were allowed to search and oviposit for 24 h. Their larvae will parasitize the host (for a longer period of time). Therefore, review and rewrite the sentence.
Lines 144-145 – “...(Fig. 1A, B), it increased on a marginal probability of significance...”. For clarity I suggest to replace “it” by “brood size”.
Lines 153-154 – “...Sex ratio was 13.9% on average...”. Authors should include information on how sex ratio was determined. As sex ration could be expressed as Female: Male or even Females: (Females+Males), it is not clear the proportion adopted.
Lines 159- on (Throughout the discussion) – scientific names of insects should be followed by the author as it is presented earlier for Plutella xylostella, for example.
Line 196 – decision instead of “dicisions”
Line 313 – remove period after the question mark (“...does host growth matter?.”)

Experimental design

Authors reviewed Material & Methods and some points are better presented.

Validity of the findings

Results and Discussion were also improved after revision.

Additional comments

As it was pointed out in the first review, the findings were very interesting but the authors should consider that the experiment used a no-choice approach. Authors did explore better the literature and improved the manuscript. As suggested earlier, some minor corrections should be taken into consideration.

Reviewer 2 ·

Basic reporting

See below

Experimental design

See below

Validity of the findings

See below

Additional comments

I read through the revised version of the manuscript, however, in my opining still a fair bit of work is required for clarity. English grammar is still my major concern and need a though revision before the manuscript could be accepted for publication.
Authors studied the effects of host stage and size and parasitoid body size and oviposition sequence on the size and sex ratio of the brood resulted from various experiment. It would nice to include all the important results in abstract and systematically in the result section. I would like to remind the author that it doesn’t matter whether they find the difference significant or not but all are the results of the study. They must include the results and discuss the them with proper references.

L23-25 – please revise the sentence - Oomyzus sokolowskii brood …. larvae
L27 revise with one … over it
L25-30 – not clear how parasitizing different instar larvae affected the brood size and sex ratio.
L44, 49– please clarify and elaborate what you mean by “state” mating or oviposting or both. A reference on mating or oviposting eg. refer http://onlinelibrary.wiley.com/doi/10.1111/j.1365-2311.2012.01347.x/pdf; https://www.jstor.org/stable/5419;
L52 include some latest references –
L60 remove “accordingly”
L63 Female Oomyzus sokolowskii “show” not “shows”
L65-67; 69-72– revise the sentence
L88-94 revise - Colony of O. sokolowskii was maintained on 4th instar P. xylostella larvae… revise
Any food given to parasitoid? If so please include that information
4 ml tube looks very small , recheck– looks like forced parasitism
What happened after parasitism – removed the parasitized larvae or let them develop in the same tube. Please provide detail information.
L96 – Larval hosts were… ; doesn’t sound ok here. Could be e.g. - Each larva was….
L99 – replaced watched with observed … oviposition behaviour
L102 – can’t be considered as replicates. You need to use fresh adult for each replicate
L107 – please could you clarify how male and female antennae differ, or provide a reference

Results: can you organise your results a bit effects of all the tested parameters on brood sixe and sex ratio irrespective of your positive or negative results.
L 133 Plase mentioned the biggest and smallest brood size within instar 4.
Not sure if you analysed the effects of larvae size across the instar or just within each instar. It’s good to check size without instar as blocks.
L 143 Please can you discuss the exponential increase in male numbers. Include the data on effects of wasp body size on sex ratio??

Discussion
L146 what percent increase in brood size?
Revise L-150 to 156 , not clear
I am still not sure the how much variation in the size of P. xylostella larvae within instar did you find? Please include this information in the result.
Probably because female exhaust all its eggs??? Discuss with references.
L179 – what do you mean by locally abundant P. xylostella ; correct the spellings
L179 – why risk?? Why female wants to save the egg for future? Discuss with references? This is gregarious parasitoid, is there any limitation on the number of wasps could develop within a host?? Discuss.
L184 – this species undergo haplodiploidy – female could control the fertilization of eggs (male or female offspring),
L194 – discuss why 1 male with 11 female ratio, multiple mating in males but females mate once in its life??? Check the references if this is the case and discuss. Female became sperm dpeleted?? References - www.sciencedirect.com/science/article/pii/S0003347209002887; onlinelibrary.wiley.com/doi/10.1111/eea.12007/pdf

Rewrite the conclusion.
References:
Please check all the references and citations

For example - The citation of this reference is missing. Kant R, Minor MA, Trewick SA Sandanayaka WRM. 2012. Body size and fitness relation in male and female 278 Diaeretiella rapae. BioControl 57:759-766. DOI: 10.1007/s10526-012-9452-4.

---

## Round 0.3 · accepted · Accept

Thanks for all your work on the revisions. I think the manuscript is now ready for publication.